# Day 100 Recovery of Absolute Number of Inhibitory KIR2DL2 and Activating NKp30 Natural Killer Cells Predicts Survival Post-Autologous Stem Cell Transplantation in Lymphomas

**DOI:** 10.3390/biomedicines12081808

**Published:** 2024-08-09

**Authors:** Luis F. Porrata, Stephen M. Ansell, Ivana N. Micallef, Patrick B. Johnston, Jose C. Villasboas, Jonas Paludo, Urshila Durani, Svetomir N. Markovic

**Affiliations:** 1Department of Medicine, Division of Hematology, Mayo Clinic, Rochester, MN 55905, USApaludo.jonas@mayo.edu (J.P.);; 2Department of Medical Oncology, Mayo Clinic, Rochester, MN 55905, USA

**Keywords:** natural killer cells, lymphoma, autologous peripheral blood hematopoietic stem cell transplantation

## Abstract

The infusion autograft absolute number of inhibitory killer immunoglobulin-like receptor (KIR) 2DL2 and activating natural killer (NK)p30 cells are predictors of clinical outcomes in lymphoma patients undergoing autologous peripheral blood hematopoietic stem cell transplantation (APBHSCT). To assess if the long-term recovery of these NK cell subsets still holds clinical relevance, we set up to investigate their prognostic ability at day 100 post-APBHSCT. This was a retrospective single-institution study including 107 patients from our prior phase III trial who had a clinical assessment at day 100 post-APBHSCT. The median follow-up from day 100 was 168.19 months (interquartile range: 156.85–181.28 months). Patients with day 100 inhibitory KIR2DL2 < 0.08 cells/µL and activating NKp30 ≥ 0.19 cells/µL experienced superior overall survival (OS) and progression-free survival (PFS). A multivariate analysis revealed both the day 100 inhibitory KIR2DL2 [OS: HR = 1.449, 95%CI, 1.231–1.895, *p* < 0.013; and PFS: HR = 2.069, 95%CI, 1.134–3.775, *p* < 0.021] and activating NKp30 [OS: HR = 4.985, 95%CI, 2.614–9.506, *p* < 0.0001; and PFS: HR = 4.661, 95%CI, 2.598–8.393, *p* < 0.0001] were independent predictors for OS and PFS. Inhibitory KIR2DL2 and activating NKp30 NK cells at day 100 are prognostic immune biomarkers in lymphoma patients treated with APBHSCT.

## 1. Introduction

Early absolute lymphocyte count (ALC) recovery defined as an ALC at day 15 (ALC-15) ≥ 500 cells/µL post-autologous peripheral blood hematopoietic stem cell transplantation (APBHSCT) is a prognostic biomarker for survival [1]. The ALC-15 recovery directly depends on the collection and the infusion of the autograft absolute lymphocyte count (A-ALC) [2]. However, neither the A-ALC or the ALC-15 depict the specific lymphocyte subset(s) associated with the observed survival advantage post-APBHSCT. Several studies have now shown that in both allogeneic and autologous stem cell transplantation, natural killer (NK) cells play a pivotal role in clinical outcomes [3,4]. The autograft infusion of the absolute number of NK cells as well as the early recovery of the absolute number of NK cells within the first 30 days post-APBHSCT has also been associated with better clinical outcomes [5,6,7], suggesting that NK cells are one of the main ALC subsets conveying better survival post-APBHSCT. However, a limitation of these reports, as it was the ALC, was mainly the analysis of the total absolute number of NK cells. To address this lack of knowledge, our group previously published that the infusion of autograft inhibitory killer immunoglobulin-like receptor (KIR) 2DL2 and activating NKp30 NK cell receptors were predictors of clinical outcomes in lymphoma patients undergoing APBHSCT [8]. The selection of inhibitory KIR2DL2 and activating NKp30 NK cells in our previous study was because of the faster and elevated levels of recovery associated with these two NK cell subsets in the post allogeneic and autologous stem cell transplantation setting [9,10]. To assess if the long-term recovery of inhibitory KIR2DL2 and activating NKp30 NK cells still functions as an immune-biomarker for prognosis post-APBHSCT, we evaluated the inhibitory KIR2DL2 and activating NKp30 NK cell recovery as a prognostic biomarker for survival at day 100 post-APBHSCT in patients who participated in our previous phase III trial [11,12].

## 2. Materials and Methods

### 2.1. Patient Population

A total 107 lymphoma patients who had participated in our double-blind, phase III randomized trial [11] and had a clinical assessment at day 100 post-APBHSCT were included in this study. Patients included in this study were 18 years of age or older with the diagnosis of non-Hodgkin’s lymphoma (NHL). Exclusion criteria included the following: (1) bone marrow harvest to collect stem cells; (2) did not achieve chemo-sensitivity defined as partial response (PR) or complete response (CR) prior to APBHSCT; and (3) participated in any APBHSCT study not using the standard BEAM (Carmustine, Etoposide, Cytarabine, and Melphalan) conditioning regimen.

### 2.2. End Points

The primary end points of this study were the impact of day 100 recovery of the inhibitory KIR2DL2 and activating NKp30 NK cells on overall survival (OS) and progression-free survival (PFS).

### 2.3. Prognostic Factors

The prognostic factors evaluated in this study included the following: international prognostic index (IPI) at diagnosis (≥2), age at diagnosis (>60 years versus ≤60 years), lactate dehydrogenase (LDH) at diagnosis (>222 U/L), ECOG (Eastern Cooperative Oncology Group) (>1 versus ≤1), performance status at diagnosis (>1 versus ≤1), extranodal sites at diagnosis (>1 versus ≤1), stage at diagnosis (III/IV versus I/II), disease status before APBHSCT [complete response (CR) versus partial response (PR)], infused CD34+ cell count, Plerixafor use, day 100 absolute number of NK cells, day 100 inhibitory KIR2DL2 NK cells, day 100 activating NKp30 NK cells, and clinical response by day 100 positron emission tomography (PET) scan.

### 2.4. Conditioning Regimen

All patients received BEAM conditioning: Carmustine (BCNU) (300 mg/m^2^) on day −6, Etoposide (100 mg/m^2^) twice daily from days −5 to −2, Cytarabine (ARA-C) (100 mg/m^2^) twice daily from days −5 to −2, and Melphalan (140 mg/m^2^) on day −1.

### 2.5. Response and Survival

OS was measured from the date of the day 100 follow-up visit after APBHSCT to the date of death or last follow-up. PFS was measured from the date of day 100 follow-up visit from APBHSCT to the time of progression, relapse, death, or last follow-up.

PET scan reports were based on the Deauville score and Lugano Classification of response [13].

### 2.6. Day 100 Peripheral Blood Inhibitory KIR2DL2 and Activating NKp30 NK Cell Analysis

Previously frozen peripheral blood mononuclear cells (0.5–1.0 × 10^6^ cells/mL) were thawed and aliquoted into 96-well round-bottomed plates (100 mL/well). The following monoclonal antibodies were used for day 100 peripheral blood mononuclear cell immunophenotyping by flow cytometry: anti-human CD3 FITC, anti-human CD16 phycoerythrin (PE), anti-human CD56 PE for NK cells, and anti-human CD158b FITC and CD337 phycoerythrin (PE) for inhibitory KIR2DL2 and activating NKp30 NK cells (BD Pharmingen, San Jose, CA, USA). The desired antibody or antibody pool was added at 5 µL of each antibody/well. The cells and antibodies were incubated for 30 min at 4 °C and washed twice with 1 × PBS (Cellgro, Manassas, VA, USA), 0.1% BSA and 0.05% sodium azide (Sigma, St. Louis, MO, USA). Three-color flow cytometry was performed on a Guava 8-HT (Millipore, St. Louis, MO, USA) and the Incyte software application (Millipore) was utilized for data analysis. After gating on live lymphocytes, the percentage of NK cells was determined by the number of cells that were CD3-negative and CD16- and CD56-positive. The percentage of inhibitory KIR2DL2 NK cells was determined by the number of cells that were CD158b-positive and CD337-negative. The percentage of activating NKp30 NK cells was determined by the number of cells that were CD337-positive and CD158b-negative. The absolute number of inhibitory KIR2DL2 and activating NKp30 NK cells at day 100 was calculated by multiplying their percentages time the day 100 absolute lymphocyte count for inhibitory KIR2DL2 and activating NKp30 NK cells.

### 2.7. Statistical Analysis

OS and PFS were analyzed using the Kaplan–Meier approach [14]. Differences between survival curves were evaluated for statistical significance using the two-tailed log-rank test. The median follow-ups from day 100 for the entire cohort, alive, and dead patients was calculated using the inverse event of interest by the Kaplan–Meier’s method.

Univariate and multivariate analysis was performed using a Cox proportional hazard model [15]. Variables with a *p* < 0.2 in the univariate analysis were included in the multivariate analysis. We evaluated the variables in the prognostic factor section to assess their impact on OS and PFS times post-APBHSCT. The choice of the cutoff values for the day 100 inhibitory KIR2DL2 and activating NKp30 NK cells to assess survival was based on their utility as a marker for the clinically relevant binary outcome death/survival using the receiver operating characteristics (ROC) curves and area under the curve (AUC). The binary clinical outcome (death/survival) was established at 5 years from the day 100 follow-up visit after APBHSCT. Patients were classified as “alive/censored” when follow-up time was >5 years and “death/uncensored” for patients known to have died before 5 years from day 100 [16]. A *K*-fold cross-validation with *K*-values of 10 was performed to validate the results of the day 100 inhibitory KIR2DL2 and activating NKp30 NK cell cutoff values obtained by the ROC curves and AUC. Randomly chosen subsets containing 90% of the cohort were used for training, and the remaining 10% were left for testing. The cross-validation process was then repeated 10 times. Based on this analysis, a cross-validation AUC by the ROC was produced, representing the discriminating accuracy of the day 100 inhibitory KIR2DLD2 and activating NKp30 NK cells for the binary clinical outcomes of death/survival.

Chi-squared tests and Fisher’s exact tests were used to determine the relationships between categorical variables, as appropriate. The Wilcoxon rank test was used to determine associations between continuous variables and categories. All *p*-values represented were two-sided, and statistical significance was declared at *p* < 0.05. The statistical software used was JMP, version pro-17, SAS institute INC, Cary, NC, USA, 1989–2023.

## 3. Results

### 3.1. Patient Characteristics

The median age of the 107 patients included in this study at the time of APBHSCT was 58 years (interquartile range (IQR): 48–62 years). The median follow-up from the day 100 visit for the entire cohort was 118.6 months (IQR: 156.85–181.28 months). For patients who were alive (N =48), the median follow-up was 166.5 months (IQR: 149.41–180.85 months), and for those who died (N = 59), the median follow-up was 49.7 months (IQR: 19.84–100.27 months) after day 100. The causes of death included the following: 43 patients of lymphoma; 4 patients with myocardial infarction; 3 patients with transplant-related acute myelogenous leukemia; 2 patients of pneumonia; one patient of anaplastic astrocytoma; one patient with transplant-related chronic obstructive pulmonary disease; 1 patient of chronic myelomonocytic leukemia; 1 patient of colon cancer; 1 patient of heart failure; 1 patient of septic shock; and 1 patient of prostate cancer.

### 3.2. Cutoff Values for Day 100 Inhibitory KIR2DL2 and Activating NKp30 NK Cells

ROC curves and AUC were used to determine the optimal cutoff values for the day 100 inhibitory KIR2DL2 and day 100 activating NKp30 NK cells. The best cutoff value for day 100 inhibitory KIR2DL2 NK cells was 0.08 cells/µL, for day 100 activating NKp30 NK cells was 0.19 cells/µL (see Appendix A) The internal validation using the *k*-fold cross validation supported the same cut-off values for the day 100 inhibitory KIR2DL2 NK cells and the day 100 activating NKp30 NK cells (see Appendix A). The day 100 absolute number of NK cells used in this study was 250 cells/µL, based on our previous publication [17].

To minimize bias due to the retrospective nature of the study, we compared the following patient baseline characteristics: patients with day 100 inhibitory KIR2DL2 < 0.08 cells/µL versus day 100 inhibitory KIR2DL2 ≥ 0.08 cells/ul; patients with day 100 activating NKp 30 < 0.19 cells/µL versus day 100 activating NKp30 ≥ 0.19 cells/µL NK cells. In the day 100 inhibitory KIR2DL2 group, both groups were balanced except for stage and IPI score. In the day 100 activating NKp30 group, both groups were balanced except for LDH, IPI score, IPI, and day 100 PET scan results (see Table 1).

### 3.3. Predictors for OS and PFS

Based on the univariate Cox regression analysis, the following variables were predictors for OS: age, LDH, extranodal disease, IPI, PR vs. CR prior to transplant, infusion of CD34, Plerixafor, day 100 PET scan, day 100 absolute number of NK cells, day 100 inhibitory KIR2DL2 NK cells, and day 100 activating NKp30 NK cells. Predictors for PFS in the univariate analysis included the following: stage, PR vs. CR prior to transplant, day 100 PET scan, day 100 absolute number of NK cells, day 100 inhibitory KIR2DL2 NK cells, and day 100 activating NKp30 NK cells (Table 2).

Table 3 shows that day 100 inhibitory KIR2DL2 NK cells [OS: HR = 1.449, 95%CI, 1.231–1.895, *p* < 0.013; and PFS: HR = 2.069, 95%CI, 1.134–3.775, *p* < 0.021] and activating NKp30 NK cells [OS: HR = 4.985, 95%CI, 2.614–9.506, *p* < 0.0001; and PFS: HR = 4.661, 95%CI, 2.598–8.393, *p* < 0.0001] were independent predictors for OS and PFS in the multivariable analysis.

### 3.4. Survival Outcomes Based on Day 100 Inhibitory KIR2DL2 and Day 100 Activating NKp30 NK Cells

Using the Kaplan–Meier method, we evaluated if there was any difference in OS and PFS comparing patient with day 100 inhibitory KIR2DL2 NK cells < 0.08 cells/µL versus patients with day 100 inhibitory KIR2DL2 NK cells ≥ 0.08 cells/µL. We observed a superior OS (Figure 1A) and PFS (Figure 1B) in patients with day 100 inhibitory KIR2DL2 NK cells < 0.08 cells/µL compared with patients with day 100 inhibitory KIR2DL2 NK cells ≥ 0.08 cells/µL: Median OS was not reached versus 100.3 months, and 5-year OS rates were 65% (95%CI, 50–77%) versus 44% (95%CI, 32–57%), *p* < 0.016, respectively; the median PFS was 180.4 months versus 36 months, and 5-year PFS rates were 61% (95%CI, 46–74%) versus 28% (95%CI, 18–41%), *p* < 0.0001, respectively. We also observed a superior OS (Figure 1C) and PFS (Figure 1D) in patients with day 100 activating NKp30 NK cells ≥ 0.19 cells/µL compared with patients with day 100 activating NKp30 NK cells < 0.19 cells/µL: the median OS was not reached versus 53 months, and 5-year OS rates were 81% (95%CI, 69–90%) versus 21% (95%CI, 12–35%), *p* < 0.0001, respectively; the median PFS was not reached versus 23.5 months, and 5-year PFS rates were 71% (95%CI, 58–82%) versus 8% (95%CI, 3–19%), *p* < 0.0001, respectively.

### 3.5. Influence of the Inhibitory KIR2DL2 and Activating NKp30 NK Cells on Survival in Subgroup Analysis

We performed a subgroup analysis to identify which patients would potentially benefit from the day 100 recovery of the inhibitory KIR2DL2 and activating NKp30 NK cells. Figure 2 shows forest plots for OS for day 100 inhibitory KIR2DL2 NK cells. The forest plots for PFS day 100 inhibitory KIR2DL2 NK cells and OS and PFS for day 100 activating NKp30 NK cells are in the Appendix A. The day 100 recovery of both inhibitory KIR2DL2 and activating NKp30 NK cells were discriminators for OS and PFS for the following: age > 60 or ≤60 years, extranodal disease > 1 of ≤1, IPI > 2 or ≤2, normal or abnormal LDH, performance status > 1 or ≤1, stage I/II or III/IV, disease status prior to APBHSCT complete response or partial response; day 100 PET scan.

## 4. Discussion

The primary objective of this study was to assess the prognostic ability of long-term inhibitory KIR2DL2 and activating NKp30 NK cells at day 100 post-APBHSCT on clinical outcomes in lymphoma patients.

We identified that both inhibitory KIR2DL2 and activating NKp30 NK cells were independent predictors for OS and PFS. Patients with day 100 inhibitory KIR2DL2 < 0.08 cells/µL and activating NKp30 ≥ 0.19 cells/µL NK cells experienced better OS and PFS compared with those who did not. Furthermore, both day 100 inhibitory KIR2DL2 and activating NKp30 NK cells were independent of the day 100 absolute number of NK cells, suggesting these two types of NK cell subsets were associated with the survival benefit observed post-APBHSCT. 

In the stem cell transplantation setting, the quantitative recovery and qualitative functional recovery of NK cells are important predictors for survival. In regard to the quantitative recovery of NK cells, both in the allogeneic and autologous stem cell transplantation, the infusion of a higher NK cell graft content has been associated with superior survival. In allogeneic stem cell transplantation, rapid and sustained NK cell reconstitution from 28 days and up to 2 years after allogeneic stem cell transplantation is associated with better survival. In the APBHSCT, the reconstitution of the absolute number of NK cells from day 15 up to day 100 post-APBHSCT also predicts better survival [6,17]. However, to our knowledge, the long-term day 100 quantitative recovery of NK cell subsets such as KIR2DL2 and NKp30 has not been previously reported in APBHSCT. Our findings argue in favor of the possible role of NK cells in cancer immunosurveillance post-APBHSCT. What biological factors directly contribute to the long-term qualitative and quantitative recovery of NK cells post-APBHSCT are currently unknown. The mechanism of action of how NK cells target cancer cells has been well described previously [18]. The graft-versus-tumor effect observed by NK cells in the allogeneic stem cell transplantation depends on the mismatch between the donor NK cell KIR expression [19] and the human leukocyte antigen (HLA) I ligands on the recipient cancer cells. Clinical outcomes based on the KIR-HLA receptor–ligand mismatch in lymphoma [20], acute myelogenous leukemia [21], and multiple myeloma [22] have been reported in autologous stem cell transplantation. Thus, a better understanding of how the KIR-HLA receptor–ligand mismatch improves survival in APBHSCT and would provide another mechanism of action of targeting cancer cells, as opposed to only relying on the high-dose chemotherapy in APBHSCT. Different NK cell strategic modalities have been used in the stem cell transplantation setting. One modality is the cytokine-based ex vivo activation, expansion, and infusion of NK cells post-APBHSCT. In multiple myeloma, the infusion of ex vivo-activated and -expanded autologous NK cells resulted in a measurable response by the decreased M-component in addition to minimal residual disease [23]. Interleukin-15 (IL-15) is a crucial cytokine for sustained proliferation and activation of NK cells. By day 15 post-APBHSCT, higher levels of plasma interleukin (IL)-15 have been associated with a higher NK cell recovery and better survival [24]. Similar findings have been observed in the allogeneic stem cell transplantation. However, IL-15 at 6 months post-allogeneic stem cell transplantation returned to pre-transplantation levels, correlating with decreased expansion of anti-tumor NK cell receptors NKp30, NKp46, and NKG2D [10]. Another NK cell treatment modality has been the use of IL-15 post-stem cell transplant. In a phase I study, the use of the super-agonist ALT-803 (IL-15) has been shown to stimulate the activation and proliferation of NK cells [25]. Combining other immunotherapy modalities with APBHSCT has shown overall improved clinical outcomes. Chimeric antigen receptor-T cell (CART) therapy has been combing with APBHSCT. Studies have been conducted on patients receiving sequential CD19/CD20/CD22 CAR-T cell therapy post-APBHSCT in lymphoma, showing good clinical outcomes [26]. Currently, several trials are underway using CART-NK-based therapy [27]. Understanding the specific NK subsets affecting survival in APBHSCT could provide a platform to develop, for example, targeted CART-NK cell therapy to combine with APBHSCT.

Limitations of our study includes the following: (1) a retrospective study design and (2) only evaluating day 100 inhibitory KIR2DL2 and day 100 activating NKp30 NK cells as other NK cell subsets could also affect survival post-APBHSCT. The strength of the study includes (1) a long-term follow-up of 13 years and (2) forest plots depicted the discriminatory ability of both day 100 inhibitory KIR2DL2 and day 100 activating NKp30 NK cells based on patient baseline characteristics.

## 5. Conclusions

In summary, this retrospective study showed the importance of the day 100 inhibitory KIR2DL2 and activating NKp30 NK cells with regard to understanding how the host immunity (NK cells) could improve survival in lymphoma patients treated with APBHSCT.

## Figures and Tables

**Figure 1 biomedicines-12-01808-f001:**
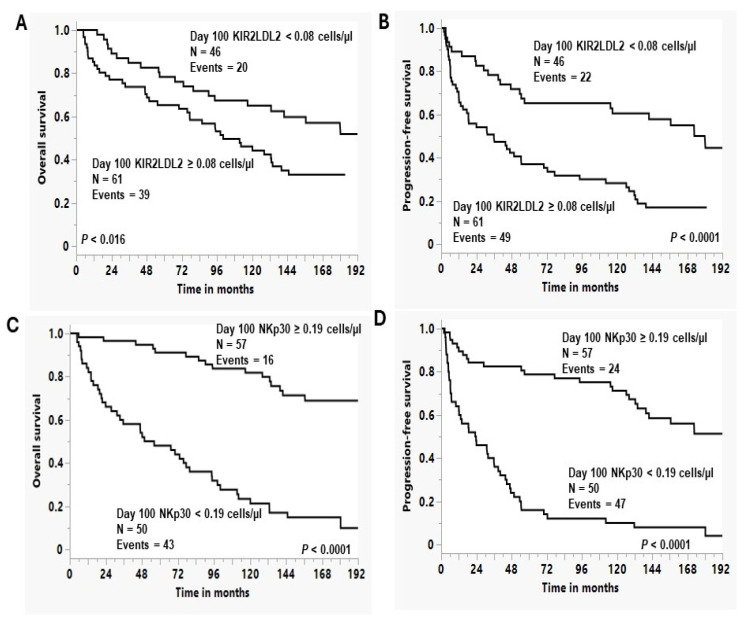
(**A**) Overall survival (OS) based on day 100 KIR2DL2 NK cells; (**B**) progression-free survival (PFS) based on day 100 KIR2DL2 NK cells; (**C**) overall survival (OS) based on day 100 NKp30; (**D**) progression-free survival (PFS) based on day 100 NKp30 NK cells.

**Figure 2 biomedicines-12-01808-f002:**
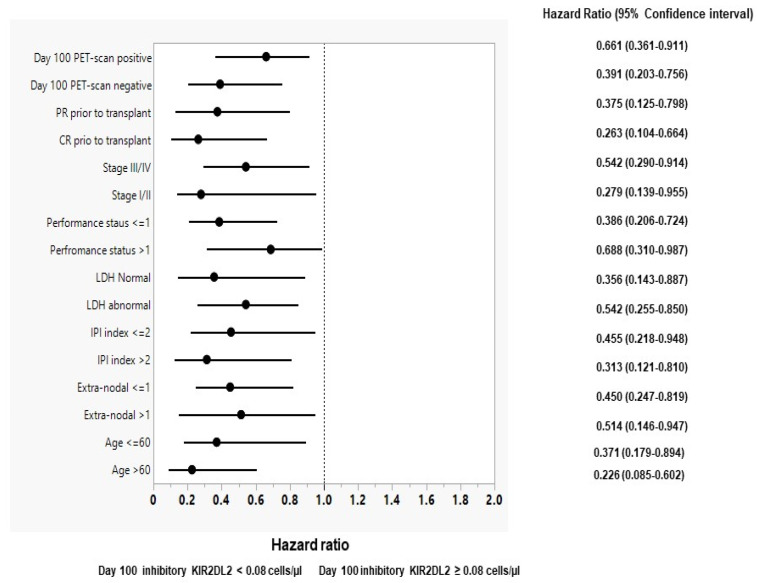
Forest plot for subgroup analysis for overall survival for day 100 inhibitory KIR2DL2 < 0.08 cells/µL versus day 100 inhibitory KIR2DL2 ≥ 0.08 cells/µL NK cells.

**Table 1 biomedicines-12-01808-t001:** Baseline characteristics of patients according to the day 100 peripheral blood absolute number of inhibitory KIR2DLD2 and activating NKp30 natural killer cell receptors.

Characteristics	KIR2LDL2 < 0.08 Cells/µ/ (N = 46)	KIR2LDL2 ≥ 0.08 Cells/µ/ (N = 61)	*p*-Value	NKp30 < 0.19 Cells/µ/ (N = 50)	NKp30 ≥ 0.19 Cells/µ/ (N = 57)	*p*-Values
** *At diagnosis* **						
Age in years, median (IQR)	56 (46–63)	55 (49–61.5)	0.623	57 (49.8–63)	55 (46.5–61)	0.252
Gender			0.642			0.818
Female	9 (20%)	15 (25%)		12 (24%)	12 (21%)	
Male	37 (80%)	46 (75%)		38 (76%)	45 (79%)	
LDH (U/L), median (IQR)	209.5 (178.5–330.5)	200 (170–298)	0.228	273 (183.5–310.3)	189 (170–270)	<0.042
Stage			<0.003			0.324
I	7 (15%)	0 (0%)		1 (2%)	6 (11%)	
II	6 (13%)	4 (7%)		5 (10%	5 (9%)	
III	7 (15%)	20 (33%)		12 (24%)	15 (26%)	
IV	26 (57%)	37 (60%)		32 (64%)	31 (54%)	
Extranodal disease			0.399			0.177
0	24 (52%)	31 (51%)		24 (48%)	31 (54%)	
1	21 (46%)	25 (41%)		21 (42%)	25 (44%)	
2	1 (2%)	5 (8%)		5 (10%)	1 (2%)	
Performance status			0.296			0.057
0	14 (30%)	11 (18%)		7 (14%)	18 (32%)	
1	28 (61%)	42 (69%)		35 (70%)	35 (61%)	
2	4 (9%)	8 (13%)		8 (16%)	4 (7%)	
IPI score			<0.019			<0.036
0	10 (22%)	5 (8%)		4 (8%)	11 (19%)	
1	15 (33%)	17 (28%)		12 (24%)	20 (35%)	
2	8 (17%)	26 (42%)		17 (34%)	17 (30%)	
3	12 (26%)	9 (15%)		12 (24%)	9 (16%)	
4	1 (2%)	4 (7%)		5 (10%)	0 (0%)	
Histology			0.409			0.415
DLBCL	26 (57%)	25 (41%)		28 (56%)	23 (40%)	
Follicular lymphoma	5 (11%)	10 (16%)		6 (12%)	9 (16%)	
Mantle cell lymphoma	10 (21%)	15 (25%)		9 (18%)	16 (28%)	
Other	5 (11%)	11 (18%)		7 (145)	9 (16%)	
** *IPI risk factors* **						
Age in years			0.536			0.217
>60	13 (28%)	21 (34%)		19 (38%)	15 (26%)	
≤60	33 (72%)	40 (66%)		31 (62%)	42 (74%)	
LDH (U/L)			0.695			<0.012
>222	22 (48%)	26 (43%)		29 (58%)	15 (33%)	
≤222	24 (52%)	35 (57%)		21 (42%)	42 (74%)	
Extranodal disease			0.234			0.096
>1	1 (2%)	5 (8%)		5 (10%)	1 (2%)	
≤1	45 (98%)	56 (92%)		45 (90%)	56 (98%)	
Performance status			0.549			0.219
>1	4 (9%)	8 (13%)		8 (16%)	4 (7%)	
≤1	42 (91%)	53 (87%)		42 (84%)	53 (93%)	
Stage			<0.029			0.428
I/II	13 (28%)	4 (7%)		6 (12%)	11 (19%)	
III/IV	33 (72%)	57 (93%)		44 (88%)	46 (81%)	
IPI			0.496			<0.041
>2	13 (28%)	13 (21%)		17 (34%)	9 (16%)	
≤2	33 (72%)	48 (79%)		33 (66%)	48 (84%)	
** *At Transplant* **						
Clinical status			0.556			0.083
CR	28 (61%)	33 (54%)		24 (48%)	37 (65%)	
PR	18 (39%)	28 (46%		26 (52%	20 (35%)	
Plerixafor			0.426			0.554
Yes	16 (35%)	27 (44%)		24 (48%)	21 (37%)	
No	30 (65%)	34 (56%)		26 (52%)	36 (63%)	
Infused CD34 cells/kg; medina (IQR)	5.13 (4.48–6.01)	5.06 (4.39–5.57)	0.287	5.26 (4.42–5.58)	4.93 (4.48–5.78)	0.443
** *At day 100 post-transplant* **						
PET scan			0.806			<0.001
Negative	38 (83%)	48 (79%)		53 (93%)	33 (66%)	
Positive	8 (17%)	13 (21%)		4 (7%)	17 (34%)	

Abbreviations: IPI = International Prognostic Index; IQR = interquartile range; LDH = lactate dehydrogenase; PET scan = positron emission tomography.

**Table 2 biomedicines-12-01808-t002:** Univariate analysis for overall survival and progression-free survival.

Variables	Overall Survival			Progression-Free Survival		
	HR	95%CI	*p*	HR	95%CI	*p*
Age > 60 years	2.130	1.273–3.564	<0.005	1.310	0.810–2.117	0.277
Female versus male	0.987	0.533–1.828	0.968	0.930	0.526–1.645	0.796
LDH > 222 U/L	1.942	1.162–3.245	<0.011	1.366	0.856–2.178	0.193
Stage III/IV	1.978	0.850–4.607	0.114	2.489	1.137–5.438	<0.022
Extranodal disease > 1	3.492	1.383–8.813	<0.008	2.396	0.960–5.981	0.061
Performance status > 1	1.549	0.735–3.268	0.250	1.602	0.820–3.129	0.168
IPI > 2	2.672	1.572–4.544	<0.0003	1.769	1.067–2.932	<0.027
PR vs. CR prior to transplant	1.870	1.116–3.131	<0.017	2.079	1.297–3.331	<0.002
CD34 numbers (continuous)	0.708	0.582–0.854	<0.0004	0.817	0.693–0.955	<0.014
Plerixafor No vs. yes	1.830	1.092–3.069	<0.023	1.564	0.974–2.511	0.064
Day 100 PET scan (positive vs. negative)	2.731	1.532–4.870	<0.0007	2.259	1.306–3.906	<0.004
Day 100 NK cells < 250 cells/µL	4.084	2.333–7.147	<0.0001	3.357	2.065–5.459	<0.0001
Day 100 KIR2LD2 > 0.08 cells/µL	1.911	1.111–3.286	<0.016	2.764	1.654–4.619	<0.0001
Day100 NKp30 < 0.19 cells/µL	6.130	3.412–11.00	<0.0001	5.332	3.183–8.932	<0.0001

Abbreviations: CR = complete response; IPI = International Prognostic Index; NK cells = natural killer cells; PET scan = positron emission tomography; PR = partial response.

**Table 3 biomedicines-12-01808-t003:** Multivariate analysis for overall survival and progression-free survival.

Variables	Overall Survival			Progression-Free Survival		
	HR	95%CI	*p*	HR	95%CI	*p*
Age > 60 years	1.402	0.702–2.799	0.343			
LDH > 222 U/L	1.122	0.470–2.678	0.846			
Stage III/IV	1.006	0.380–2.598	0.914	1.810	0.790–4.151	0.268
Extranodal disease > 1	1.690	0.501–5.709	0.422	1.489	0.485–4.570	0.547
Performance status > 1	1.549	0.735–3.268	0.343			
IPI > 2	1.324	0.426–4.116	0.649	1.324	0.695–2.523	0.417
PR vs. CR prior to transplant	1.708	0.921–3.166	0.093	2.511	1.434–4.397	<0.001
CD34 numbers (continuous)	0.729	0.364–1.457	0.427	0.827	0.667–1.012	0.073
Plerixafor No vs. yes	1.142	0.597–2.187	0.722	1.277	0.712–2.279	0.422
Day 100 PET scan (positive vs. negative)	1.055	0.527–2.110	0.912	2.018	0.570–1.821	0.9
Day 100 NK cells < 250 cells/µL	2.610	1.414–4.816	<0.002	2.484	1.444–4.292	<0.001
Day 100 KIR2LD2 > 0.08 cells/µL	1.449	1.231–1.895	<0.013	2.069	1.134–3.775	<0.021
Day100 NKp30 < 0.19 cells/µL	4.985	2.614–9.506	<0.0001	4.661	2.598–8.393	<0.0001

Abbreviations: CR = complete response; IPI = International Prognostic Index; NK cells = natural killer cells; PET scan = positron emission tomography; and PR = partial response.

## Data Availability

The datasets generated during and/or analyzed during the current study are available from the corresponding author on reasonable request.

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
