# Peer review of "Day 100 Recovery of Absolute Number of Inhibitory KIR2DL2 and Activating NKp30 Natural Killer Cells Predicts Survival Post-Autologous Stem Cell Transplantation in Lymphomas"

_biomedicines, 2024, doi:10.3390/biomedicines12081808_

Round 1

Reviewer 1 Report

Comments and Suggestions for Authors

In this manuscript, authors presented Day 100 recovery of absolute numbers of the inhibitory KIR2DL2 and activating NKp30 Natural Killer Cells Predicts Survival Post-Autologous Stem Cell Transplantation in Lymphomas.

In my opinion, the topic is original and has a strong scientific value.

There are a few points that need to be addressed before the article gets accepted:

1. All abbreviations should be explained every first time they appear in text or abstract.

2. Improve the Introduction. Provide more references for this part.

3. Plagiarism is above 40%, reduce it below 10%.

4. Try reducing self-citation by less than 15% or adding more new references.

5. The cell densities used in the cell analysis are unusually high range for 96 well plates. Can you explain?

6. The conclusion part needs to be improved.

I will be happy to recommend the publication of this paper after minor corrections.

Author Response

See attach response letter

Reviewer 2 Report

Comments and Suggestions for Authors

Dear Authors,

Your article is very interesting and reveals important information for lymphoma patients’ treatment.

However I have several comments.

The introduction is too short. I would suggest to widen it.

Materials and Methods: page 3, line 120: Please name these non-parametric tests that were used. In fact I did not see any results evaluating the associations between continuous variables.

Also please add the name of the statistical software as well as its version.

Results: page 3, line 127-127: please remove these sentences.

I would suggest using interquartile range instead of the ordinary range for continuous variables.

The median follow up could also be calculated by Kaplan-Meier’s method as using the inverse event of interest.

The p-values should have 3 decimals. This is valid also for non-significant ones. Please take a look at Table 3: the last 3 p-values for OS are the following: <0.002; <0.01; <0.0001. These numbers show that the Authors prefer the last predictor because its significance has 4 decimals. The least prefer predictor by the Authors is the second one because it has only 2 decimals. In fact the least meaningful p-value is <0.001. Please correct all p-values in the text, the tables, and the figures. If you wish to insist that a given factor’s p-value is sooo small, you could highlight it in the text (for example p<0.0001) but please do not cut a p-value’s third digit (a bad example is p<0.01), it is better to write the number instead. I could guess that p<0.01 is in fact p=0.009.

In addition when all p-values have 3 decimals and they are aligned to the right, it is very easy for the readers to assume the information and to assess which factor is significant.

No supplementary materials were presented for review.

Author Response

See attach response letter
